# Vitamin interdependencies predicted by metagenomics-informed network analyses and validated in microbial community microcosms

Tomas Hessler[1,2,3], Robert J. Huddy [4], Rohan Sachdeva[1,2], Shufei Lei [2], Susan T. L. Harrison [5,6,7], Spencer Diamond [1,2] & Jillian F. Banfield [1,2,8] ✉

Metagenomic or metabarcoding data are often used to predict microbial interactions in complex communities, but these predictions are rarely explored experimentally. Here, we use an organism abundance correlation network to investigate factors that control community organization in mine tailings-derived laboratory microbial consortia grown under dozens of conditions. The network is overlaid with metagenomic information about functional capacities to generate testable hypotheses. We develop a metric to predict the importance of each node within its local network environments relative to correlated vitamin auxotrophs, and predict that a *Variovorax* species is a hub as an important source of thiamine. Quantification of thiamine during the growth of *Variovorax* in minimal media show high levels of thiamine production, up to 100 mg/L. A few of the correlated thiamine auxotrophs are predicted to produce pantothenate, which we show is required for growth of *Variovorax*, supporting that a subset of vitamin-dependent interactions are mutualistic. A *Cryptococcus* yeast produces the B-vitamin pantothenate, and co-culturing with *Variovorax* leads to a 90-130-fold fitness increase for both organisms. Our study demonstrates the predictive power of metagenome-informed, microbial consortia-based network analyses for identifying microbial interactions that underpin the structure and functioning of microbial communities.

Microbial communities are critical to human health, agriculture, industrial bioprocesses and the functioning of natural environments such as soil, groundwater and oceans. However, our ability to predict the structures of microbial communities, and how these might be altered by perturbations such as sustained environmental changes, is lacking. Microbial communities are underpinned by numerous interactions, the understanding of which should enhance our ability to manipulate and improve microbiome functioning.

[1]The Innovative Genomics Institute at the University of California, Berkeley, CA, USA. [2]The Department of Earth and Planetary Science, University of California, Berkeley, CA, USA. [3]Environmental Genomics and Systems Biology Division, Lawrence Berkeley National Laboratory, Berkeley, CA, USA. [4]Reasearch Office, Faculty of Health Sciences, University of Cape Town, Cape Town, South Africa. [5]The Center for Bioprocess Engineering Research, University of Cape Town, Cape Town, South Africa. [6]The Future Water Institute, University of Cape Town, Cape Town, South Africa. [7]Department of Chemical Engineering, University of Cape Town, Cape Town, South Africa. [8]The Department of Environmental Science, Policy and Management, University of California, Berkeley, CA, USA. ✉e-mail: jbanfield@berkeley.edu

Correlation-based network analysis is one of the longest-standing methods for identifying interacting organisms in ecosystems[1]. In the context of microbial communities, such networks and interactions have, by and large, been predicted using abundance and taxonomic information inferred from 16 S rRNA gene amplicon sequencing data (as reviewed by Matchado et al.[2] and Banerjee et al.[3]). However, 16 S rRNA sequencing does not provide any information about an organism's functional potential and also suffers from the fact that the network nodes representing the quantifiable units of 16 S analysis (amplicon sequence variants or operational taxonomic units) are not necessarily synonymous with independent microbial species. Thus, networks constructed from 16 S data likely do not represent co-occurrences of unique and independent species as well as lack any functional basis to explain why some organisms are network hubs or exhibit strong interdependencies with other organisms.

To generate a mechanistic understanding of interactions in communities, Qian et al.[4] recently suggested moving away from correlation-based networks to modeling approaches. These include Generalized Lotka-Voltrerra models[5] and Dynamic Flux balance analysis models[6] that are based on genomic information. For example, genomic information has been used to generate metabolic models that predict the interactions between two species in co-culture[7] and have demonstrated that cross-feeding is common. The inherent nature of cross-feeding between organisms that do not encode for the biosynthesis of their own metabolites necessary for growth gave rise to the Black Queen hypothesis[8]. This hypothesis supposes that the loss of genes and pathways enables the conservation of an organism's resources and leads to microbial dependencies.

Despite the progress and the promising outlook of these modelling approaches, they remain difficult to implement for complex microbial communities under varied conditions, and are therefore, largely limited to simple microbial systems. In contrast, we hypothesized that the functional basis for microbial interactions can be predicted if genome-resolved metagenomic data for complex microbial communities are incorporated into conventional network analyses.

Our previous work has produced and analyzed metagenome datasets from numerous bioreactors which were inoculated with thiocyanate-degrading (SCN⁻) microbial culture originating from a mine tailings reservoir, and operated under varying conditions[9–12]. As vitamin B12 (cobalamin) is produced by only a subset of organisms within these systems[12], we hypothesize that vitamin production may underlie the functioning of specific keystone organisms in these communities. Vitamins are essential for cellular function, and vitamin auxotrophs are frequently observed within microbial communities[13,14]. Not all bacteria need certain vitamins. A vitamin auxotroph is therefore defined as an organism (i) unable to synthesize its own vitamin (ii) which it requires for growth. However, Xavier et al.[15] predicted some vitamins and cofactors as essential in bacteria, including thiamine, coenzyme-A (for which pantothenic acid is required for biosynthesis) and NAD(H). Sokolovskaya et al.[14] proposed cobamides as a model

system for studying microbial interactions, in part because their biosynthetic pathways are well annotated and are near-essential. Experimental evidence supports this focus. For example, a prior study generated vitamin, amino acid, nucleotide and carbon metabolism auxotrophs in *E. coli* and found that vitamin auxotrophs were more likely to enter into cross-feeding interactions with organisms with complementary auxotrophies in mixed communities[16].

In this study, we constructed a correlation-based network for bioreactor-grown microbial communities using previously published thiocyanate bioreactor metagenome datasets and overlaid functional information onto this network, focusing specifically on vitamin biosynthesis. Using our approach, we identified organisms that are hubs in the correlation network, including one that is predicted to supply thiamine to correlated thiamine auxotrophs. We tested this prediction by isolating this predicted thiamine producer and thiamine auxotrophs and by performing co-culture experiments in which the amount of thiamine produced was quantified. We suggest that the overlay of genome-resolved metagenomic functional information onto network analyses can more generally explain the metabolic interactions that occur within subnetworks of microbial communities.

## Results

### Microbial community network structure

This study leveraged time series, genome-resolved metagenomic datasets that were acquired from bioreactors that were inoculated with a microbial consortium originating from a single environmental community sampled from a mine tailings site, and operated over a range of conditions, including varying thiocyanate (SCN) concentrations, applied hydraulic retention times, and with or without the addition of molasses as an additional energy source[9–12]. In these reactor studies, certain operational conditions were kept constant while others were varied. The reactors themselves were identical, were all maintained at room temperature, and the same base media composition was used in all experiments. However, the incorporation of molasses, the thiocyanate concentration, and the dilution rate were varied in these studies.

Because the bioreactors contained organisms able to degrade thiocyanate, the thiocyanate concentrations remained below detection levels, preventing any toxicity effects. The 92 metagenomic datasets included draft genomes for virtually all the relatively abundant organisms. We used this data to construct an abundance-based co-occurrence network onto which we overlaid functional capacity predictions, predicted vitamin auxotrophies, and identified and tested these predicted microbial interactions (Fig. 1).

We performed correlation analysis using Fastspar[17] which implements the SparCC algorithm[18], which has been developed to address the challenges associated with compositional data. We used positive linear correlations between the 309 species-representative genomes recovered across the 92 bioreactor metagenomes (Fig. 2a, see methodology). Organisms that had strong positive correlations were included in the correlation network. This network has 120 nodes

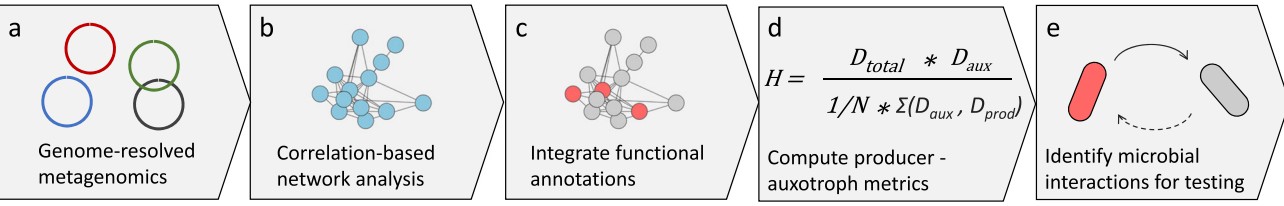

**Fig. 1 | Schematic diagram of the computational and statistical approach employed to identify microbial interactions using genome-resolved metagenomics and network analyses in this study.** First, metagenome-associated genomes are recovered and used for abundance-based correlation network analysis (**a**, **b**). Functional traits, such as vitamin biosynthesis, are inferred from the genomes and overlaid onto the network (**c**). The importance of nodes with a given

functional annotation is estimated (**d**; see Methods for detailed explanation of this equation). The generated metric estimates the importance of a vitamin-producing organism for the provisioning of this vitamin to other members in the local network neighborhood. This leads to the generation of hypotheses describing potential microbial interactions (**e**).

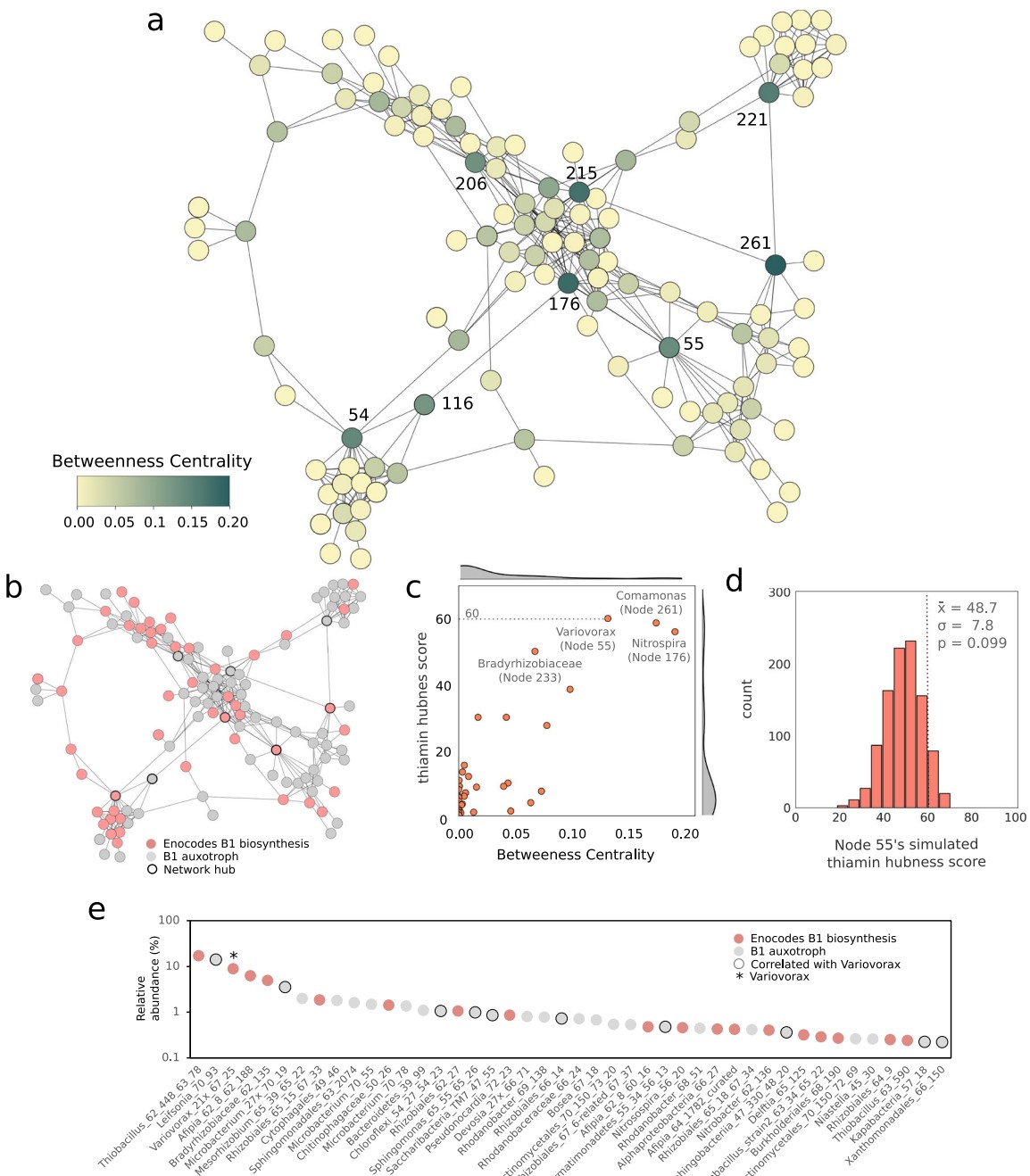

**Fig. 2 | *Variovorax* (node 55) is a network hub predicted to be important for thiamine production in the bioreactor communities. a** Network analysis based on sparCC correlation across 92 metagenomes was used to predict associations of microorganisms with other members. The nodes with the greatest betweenness centrality were delineated as network hubs and are numbered. Four of these network hubs (**b**) encode genes for the biosynthesis of thiamine (vitamin B1). Of these four hubs (**c**) *Variovorax* (Node) 55 shows the greatest importance for correlated auxotrophic members based on thiamine-hubness metrics which considers the degree, connections to B1-deficient nodes, and the number of connections those B1-deficient nodes have to B1-producers. **d** Distribution of calculated vitamin hub metric scores of Variovorax (Node 55), given the network shown in **a**, after randomly assigning 38 nodes within the network the capacity to produce thiamine, over 1000 simulations. **e** *Variovorax* is present in communities with many other bacteria able to produce thiamine but is correlated almost exclusively with thiamine auxotrophs within bioreactor communities. The rank abundance curve of the relative abundance of the 44 most prevalent microorganisms in bioreactor sample SCN18_30_10_14_R1_P is shown as an example of this phenomenon.

that represent independent microbial species, with an average of four neighbors per node. The total network density was just 6.7% (the percentage of total possible edges between the nodes in a network), indicating that the network was sparse (has a low level of connectivity). A network density of 100% would correspond to a network in which every node is connected to every other node. The network had a modularity of 0.52 (a measure between 0 and 1, quantifying the separation of nodes within a network into interconnected groups).

This relatively high modularity indicates extensive separation of groups of organisms into sub-networks and possibly indicating distinct communities[19,20].

We delineated eight nodes as network hubs (Fig. 1a) based on elevated betweenness centrality, a whole-network metric that represents the relative importance of a node for the overall connectivity within a network. The hubs were three Alphaproteobacteria (a *Mesorhizobium* sp., a *Sphingomondales* sp. and *Rhodospirillales* sp.), three

Betaproteobacteria (*Variovorax, Commamonas, Nitrospira*), a member of the Gammaproteobacteria family Rhodanobaceraceae and a member of the Bacteroidiota family Chitinophagaceae.

The network was partitioned into modules, subnetworks which make up the network, by computing the maximum modularity of the network based on Louvain heuristics (Fig. S1). Modular networks such as this have a greater number of edges within these modules than they do with other modules. Four of the modules have a single hub organism, two lack a hub organism and two contain two hub organisms.

We investigated how specific conditions may affect the network structure. We categorized the samples into biofilm or planktonic as well as environments with high or low thiocyanate concentrations. Most network hubs showed similar frequency and abundance in biofilm and planktonic samples (Figs. S2, S3). However, the *Chitinophagacaea* (Node 221) was more frequent and abundant in planktonic communities as well as high thiocyanate samples. This was also true for the other nodes in this node's module. In contrast, the module associated with the Alphaproteobacteria hub (Node 54) was more frequent and abundant in low thiocyanate samples and in planktonic communities. The other determined hub nodes showed little difference between these conditions, suggesting they have a broader ecological tolerance within these bioreactor systems.

We quantified the possible importance of vitamin producers within their local network neighborhood (Fig. 2b) by defining a node-level vitamin-hubness metric. The metric computes the product of a node's degree (number of correlations) and its number of correlations with vitamin auxotrophs, relative to the mean number of connections these auxotrophs have with vitamin producers. This identifies well-connected vitamin producers which are correlated with many auxotrophs. The second term reduces the score of producers whose auxotrophs are correlated with other producers from which they could obtain the vitamin of interest. Using this approach, we identified multiple organisms as potentially important for the provisioning of B vitamins. These included the hub organisms *Nitrospira* (Node 176), *Comamonas* (Node 261) and *Variovorax* (Node 55) for the provisioning of thiamine, *Chitinophagacaea* (Node 221) for pantothenate, as well as a *Rhodanobacter* (Node 48) for the possible provisioning of biotin and riboflavin (Figs. S4, S5). It is worth noting that the metrics used in the analysis do not indicate that the identified organisms are the only ones in their communities capable of producing the specific vitamins. Instead, the purpose is to identify potential vitamin producers that may be relied upon by auxotrophs in the community to provide these vitamins.

We performed an isolation campaign in an attempt to recover as many organisms represented in the network as possible. This led to the isolation of 43 bacterial species and single yeast species. Only one hub organism was isolated, a *Variovorax* corresponding to Node 55. The genome sequence of the isolate corresponded (99.99% ANI) with the genome from the metagenome for the hub organism. We, therefore, focused on the subnetwork for which *Variovorax* is the hub.

We wanted to determine if the reason for *Variovorax's* high thiamine hubness of 60.0 (Fig. 2c) was due to the enrichment of thiamine auxotrophs in its local network neighborhood, or due to the absence of other thiamine producers aside from *Variovorax*, or a combination of both. We randomly assigned 38 nodes as producers (the true number of nodes found to be capable of thiamine production) and 82 as auxotrophs with the ability to produce thiamine and recalculated *Variovorax's* thiamine-hubness score. We repeated this process for a total of one thousand simulations (Fig. 2d). The distribution of these scores resembles a normal distribution with an average score of 48.2 and a standard deviation of 7.9, meaning the *Variovorax* thiamine-hubness score of 60.0 was only ~1.5 standard deviations greater than the mean. Therefore, *Variovorax's* high thiamine hubness is not a result of auxotroph enrichment in its local network neighborhood, but instead being due to *Variovorax* being one of the few thiamine producers with which the thiamine auxotrophs are correlated.

*Variovorax* was present in communities with many other predicted thiamine (vitamin B1) producers, but the majority of the organisms which were correlated with *Variovorax* were thiamine auxotrophs (Fig. 2e). Interestingly, each module contained at least one metagenome-assembled genome (MAG) with a complete pathway for the biosynthesis of thiamine; most were Alphaproteobacteria or Betaproteobacteria, (Supplementary data 1, Supplementary data 2, Fig. S6). We found that 99% of the 309 genomes analyzed in this study encoded enzymes of the TCA cycle, as shown in Fig. S6. However, three Saccharibacteria genomes were an exception as they rely on their host for these genes and metabolites[21]. Additionally, we found that each genome, including the Saccharibacteria, contained a minimum of two and a median of seven enzymes that require thiamine as a cofactor[22] (Fig. S6). Many members of the communities where *Variovorax* was present were also predicted to produce thiamine (Fig. 2e). This raises the question of why *Varivorax* rather than another thiamine producer serves as the hub organism.

We employed Flashweave[19] to explore the possibility of detecting the predicted Variovorax phenotype using an alternative correlation method (Fig. S7). The resulting network exhibited a higher number of predicted correlations compared to SparCC (Fastspar) and displayed modular characteristics, consisting of six submodules with a modularity score of 0.497. Out of the 164 genomes in this network, 49 were predicted to be capable of producing thiamine. We calculated the thiamine hub metric for these nodes, and we observed that the majority of the top-scoring nodes in this network coincided with the top-scoring nodes identified in the SparCC network. These included the Comamonas (Node 261), Bradyrhizobiaceae (Node 233), Nitrospira (Node 176) and importantly, Variovorax (Node 55). Interestingly, the Commamonas (node 261) predicted to be a network hub from the SparCC network had the highest betweenness centrality of any node in the Fastweave network.

## *Variovorax* thiamine biosynthetic operon

A single thiamine-pyrophosphate (TPP) riboswitch-regulated operon contains the majority of the known thiamine biosynthetic genes within the *Variovorax* genome. Included in this operon are a thiamine permease and a TonB-dependent receptor-like protein predicted to be a thiamine active transporter (Fig. 3a). The biosynthetic genes include the *thiD* whose product is responsible for phosphorylation of the pyrimidine moiety, several genes involved in the biosynthesis of the thiazole ring and *thiE* responsible for the linking of these two moieties. The *thiC* gene encoding for the phosphomethylpyrimidine synthase, responsible for the generation of the hydroxymethyl pyrimidine, is found in a separate TPP riboswitch-regulated operon. The remainder of the genes involved in thiamine biosynthesis (genes *iscS*, *thiL*, *dxs*) are located separately in the *Variovorax* genome. Neither the standard thiamine transporter genes *thiT* (also referred to as *yuaJ*) nor any of the *ykoFEDC* genes were identified in the *Variovorax* genome, possibly due to poor transporter annotation.

Dysregulated thiamine operon mutants have been developed to increase thiamine production for biotechnological applications[23]. This requires the mutation of three genes, one of which is the thiamine-monophosphate kinase (*thiL*). The isolated *Variovorax thiL* amino acid sequence appears functional, based on a comparison with a functional *thiL* sequence analysis performed by McCulloch et al.[24]. The active site residues of our Variovorax's *thiL* are present and the additional conserved residues identified by McCulloch and co-authors were largely maintained (Supplementary data 2).

## The *Variovorax* genome encodes several capacities potentially important for interaction

The 7.9 Mbp *Variovorax* genome is predicted to encode for a highly versatile aerobic metabolism, with the capacity to utilize a variety of sugars (including trehalose and starch), lipids, and complex and

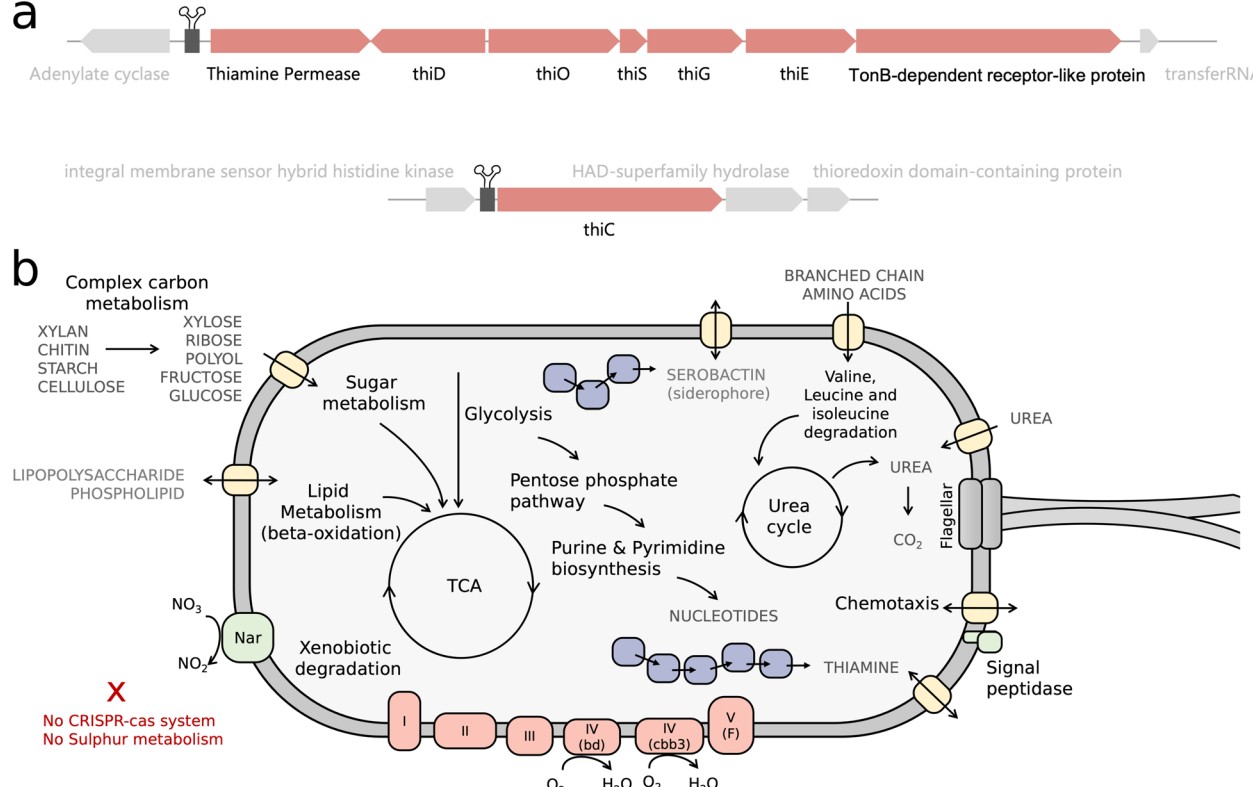

**Fig. 3 | _Variovorax_ is predicted to be capable of thiamine biosynthesis, be metabolically versatile, produce the siderophore Serobactin and has many genes involved in chemotaxis. a** Schematic diagram of the thiamine biosynthesis operons regulated by TPP riboswitches in the isolated _Variovorax_ genome, and a (**b**) cell diagram representing the genetic potential of the _Variovorax_ isolated from the bioreactor community.

recalcitrant (often plant-derived) carbon compounds such as xylan and cellulose (Fig. 3b). The genome also encodes for the biosynthesis of thiamine, the siderophore Serobactin, fatty acids, as well as encoding for a large number of genes relating to motility and chemotaxis. However, the genome lacks a complete pathway for the biosynthesis of pantothenate (vitamin B5). The _Variovorax_ isolate grew aerobically on sugars, pyruvate and branched-chain amino acids, but growth required the addition of pantothenate acid (vitamin B5). _Variovorax_ was found to be correlated with two pantothenate acid-producing bacteria, a _Sphingobacteria_ and a _Kapabacteria_.

### _Variovorax_ supports _Saccharibacteria_ and other members of enigmatic phyla

A number of bacteria correlated with _Variovorax_ belong to rare and understudied phyla including a _Chloroflexota_, a _Kapabacteria_, an _Armaimondota_ and two _Saccharibacteria_ (Fig. 4a). As is true with other _Saccharibacteria_, the _Saccharibacteria_ have small genomes and highly reduced metabolic capacities. The host of the _Saccharibacteria_ (Node 45) was previously predicted to be a _Microbacterium ginsengsoli_ (Node 144) based on co-occurrence patterns identified through hierarchical clustering[12]. Neither _Saccharibacteria_ nor _Microbacterium_ have genomes that encode for the biosynthesis of thiamine, but nonetheless, all encode several genes coding for proteins which are known to require thiamine as a cofactor. The proliferation of _Saccharibacteria_ to high abundance levels only occurred when _Variovorax_ was reasonably abundant (Fig. S8B). This suggests that _Saccharibacteria_ are dependent on both the host _Microbacterium_ and _Variovorax_.

### Thiamine biosynthesis is common in the genus _Variovorax_

_Variovorax_ has been suggested as an important hub in rhizosphere communities due to its capacity to degrade complex carbon compounds, but the findings of this study raise the possibility that vitamin production may also contribute to its hub behavior. Thus, we investigated whether thiamine production is a common trait within the _Variovorax_ genus. We assessed all _Proteobacterial_ genomes in Genome Taxonomy Database (GTDB) for their capacity to produce thiamine. The capacity to synthesise thiamine was observed in 24.5% of Gammaproteobacteria genera, 20% of Proteobacterial genera and only 10% of all 9884 Proteobacteria genomes. _Variovorax_ is among the Proteobacteria genera with the highest proportion of members that encode thiamin biosynthesis (Fig. 5b). Thiamine biosynthesis is also common among genomes classified to _Shewanella_ and _Pseudomonas_. However, members of the _Rhizobium_, _Mesorhizobium_, and _Sphingomonas_ genera appear to rarely encode for thiamine biosynthesis.

Of the 72 isolate and metagenome-derived _Variovorax_ genomes in GTDB, 27 encoded a near-complete to complete thiamine biosynthetic pathway (37.5 %). Each of these 27 genomes encoded a thiazole synthase (_thiG_), 17 of which encode a complete thiamine biosynthesis pathway and the other 10 encode a near-complete pathway (Fig. 5), as defined by the presence of genes in four major enzymatic steps (described in the methodology). In contrast, only five were found to encode a complete pantothenate biosynthesis pathway.

We investigated our bioreactor genomes and found those that encode for five or more of the vitamins and cofactors thiamine, biotin, tetrahydrofolate, riboflavin, NAD and pantothenic acid had a significantly larger genome size than those which encoded for fewer than 5 (Fig. 5c).

### _Variovorax_ increases the fitness of thiamine auxotrophs through thiamine production

To test the hypothesis that _Variovorax_ can act as a thiamine supplier to thiamine auxotrophs we screened our bioreactor isolates for microorganisms whose growth was greatly enhanced by the incorporation of thiamine into minimal media, and could therefore be used in a

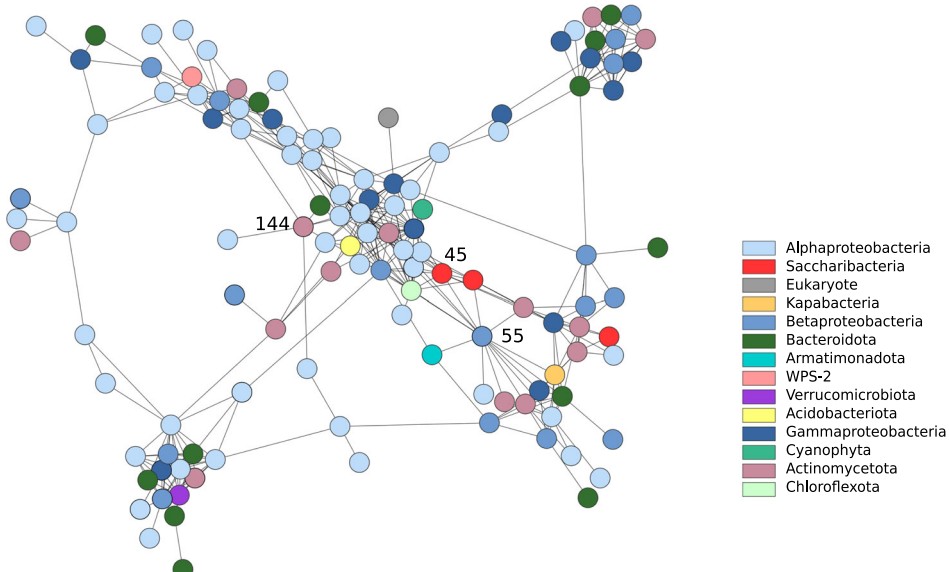

Alphaproteobacteria
Saccharibacteria
Eukaryote
Kapabacteria
Betaproteobacteria
Bacteroidota
Armatimonadota
WPS-2
Verrucomicrobiota
Acidobacteriota
Gammaproteobacteria
Cyanophyta
Actinomycetota
Chloroflexota

**Fig. 4 | *Variovorax* (Node 55) is correlated with a *Saccharibacteria* (Node 45) as illustrated in the network plot showing phyla annotations.** *Microbacterium* (Node 144) and *Saccharibacteria* both have multiple genes which encode proteins that require thiamine as a cofactor - neither of these genomes encodes for thiamine biosynthesis. A version of this figure with clearer node separation can be found in the supplementary material (Fig. S8).

bioassay. This identified yeast isolate belonging to the genus *Cryptococcus* as a thiamine auxotroph (Fig. 6a). We hypothesized that in co-culture in minimal media, the predicted thiamine producer *Variovorax* would confer a fitness advantage to the *Cryptococcus*. Indeed, the cell density of *Cryptococcus* increased ~90-fold in a co-culture (Fig. 6b). *Variovorax* too benefitted from this interaction, increasing from $7 \times 10^5$ cells/ml in pure culture to $9 \times 10^8$ cells/ml in co-culture with *Cryptococcus* (~130-fold).

To confirm that Variovorax was producing thiamine we monitored for thiamine while culturing *Variovorax* in minimal media with the addition of 25% spent media from a *Cryptococcus* five-day monoculture in minimal media. In minimal media without the addition of spent media, no visible growth of *Variovorax* was observed. The spent media enabled *Variovorax* to reach an optical density (OD$_{600}$) of ~0.15 after 96 h (Fig. 6c), but in an almost linear fashion. The thiamine concentration increased simultaneously with *Variovorax* growth from 3.6 mg/L at the start of the experiment to a maximum of 100 mg/L after 96 h.

We previously predicted *Variovorax* to be a pantothenate auxotroph based on its genome and confirmed this in a series of short growth experiments (Supplementary data 3). Despite the poor growth of the *Cryptococcus* in minimal media over five days, we were able to detect ~5 mg/L of pantothenate in its spent media, suggesting that *Cryptococcus* production of pantothenate may account for *Varioovrax's* improved growth in the co-culture. We find Variovorax to connections to two pantothenate producers in the network (Fig. S5).

### *Variovorax* produces thiamine during lag phase

We performed growth experiments of *Variovorax* in minimal media in the presence of increasing pantothenate concentrations (Fig. 7). In all instances, slow linear growth was initially observed followed by exponential growth after an extended lag phase (Fig. 7a). Increasing pantothenate concentrations allowed greater optical densities to be reached by the onset of this lag phase. Subsequently, extracellular thiamine production proceeded linearly (Fig. 7b), until the start of exponential growth, upon which the produced thiamine was rapidly consumed. Higher concentrations of supplied pantothenate corresponded with greater produced thiamine concentrations, with a maximum of 39.5 mg/L thiamine observed in the 5 mg/L pantothenate

supplemented cultured. The higher concentrations of pantothenate also allowed exponential growth to be reached sooner after inoculation. However, the specific thiamine production rate (thiamine produced per unit optical density) was largely consistent between supplied pantothenate conditions. Interestingly, *Variovorax* showed very little growth within the first 144 h in the absence of supplied pantothenate, yet still produced considerable quantities of extracellular thiamine (8.4 mg/L) over the first 144 h.

### Discussion

Here by overlaying functional metagenome information onto co-occurrence networks we identified several microorganisms in a complex microbial community which we predict to be important for the provisioning of B vitamins. We developed a hubness metric that can be used to evaluate the extent to which a resource of interest is important for structuring microbial interactions within subnetworks of networks that describe microbial community co-occurrence patterns (Fig. 2c). We used this approach to calculate the thiamine hubness metric and predicted that a *Variovorax* sp. is a hub (Fig. 2a) because it supplies thiamine to other organisms in its immediate network vicinity. The metric considers the number of connections between a hub organism (such as *Variovorax*) and other organisms in its immediate network neighborhood, the number of thiamine autotrophs in this neighborhood set, but it also considers the number of connections between these thiamine auxotrophs and predicted thiamine producers other than *Variovorax*. This last term downplays the importance of a predicted interaction in cases where other organisms (within the immediate network neighborhood or outside of it) could be alternative sources of the resource. This metric can be readily applied to future analyses to assess auxotrophies and metabolic exchange in other microbial communities. In future work, it would be interesting to explore how higher concentrations of molasses, other complex substrates, or vitamin additives would reconfigure the network.

We isolated a network hub organism *Variovorax*, which we predicted to be important for the biosynthesis of thiamine. Thiamine is a cofactor required by enzymes throughout central metabolism including those of the tricarboxylic acid cycle[25], pentose phosphate pathway[26], and amino acid catabolism and anabolism[27], yet not all

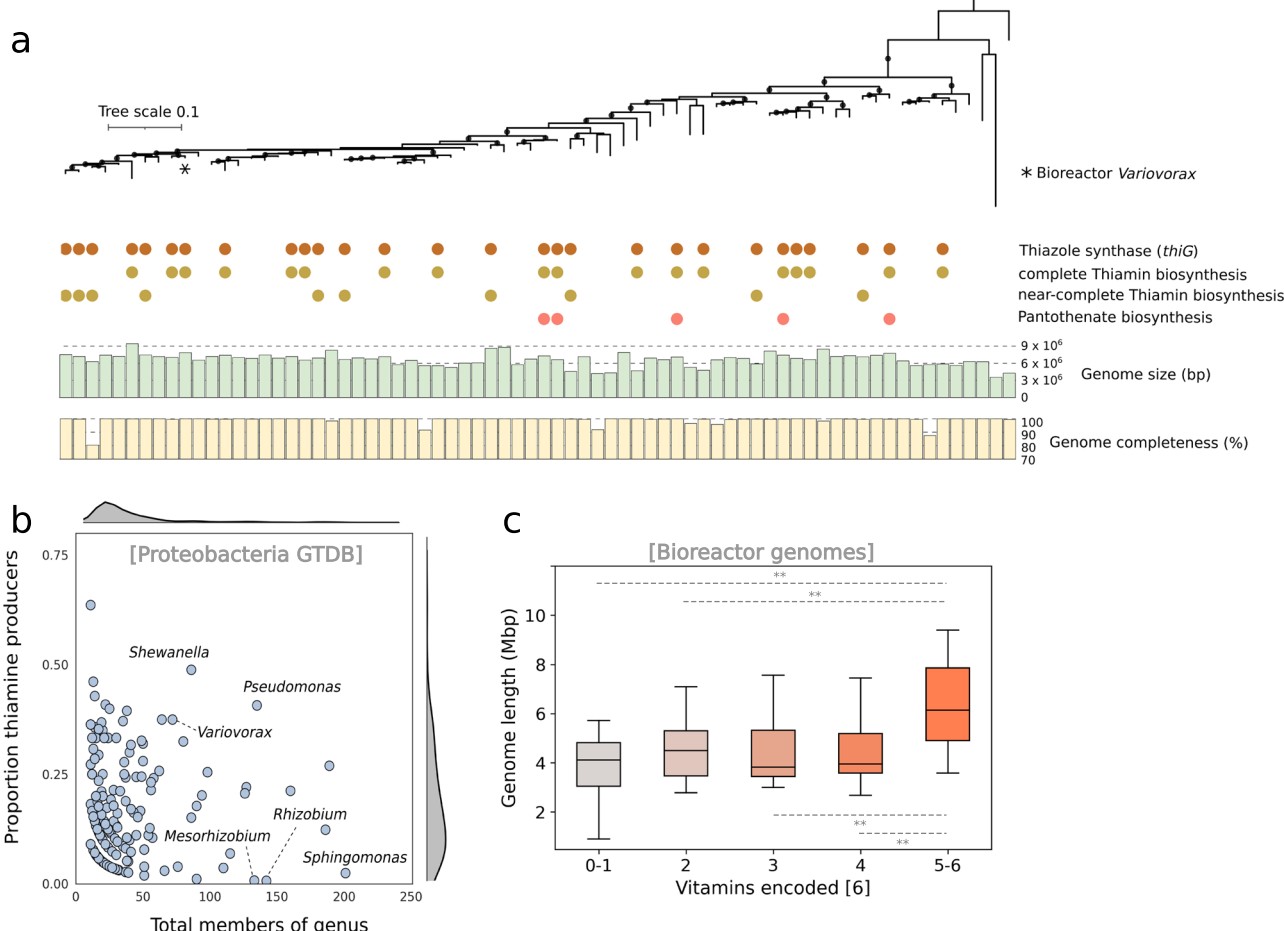

**Fig. 5 | Thiamine biosynthesis is common in *Variovorax* genomes.**
**a** Phylogenetic tree of 16 concatenated ribosomal proteins from 70 *Variovorax* genomes deposited in the Genome Taxonomy Database (GTDB) together with the *Variovorax* isolated from our bioreactor. The encoding of a complete Vitamin B1 biosynthetic pathway is shown and was found to be encoded in 27 of the 72 GTDB *Variovorax* genomes. Only five of the genomes encode a complete pantothenate biosynthetic pathway. *Rhodoferax saidenbachensis* is shown as the outgroup. Branches supported by bootstrap values greater than 75 are shown as circles. The bioreactor *Variovorax* shares the greatest similarity with *Variovorax* sp003019815 (GCF_003019815.1). **b** The GTDB genomes of genera within Proteobacteria genomes in GTDB were assessed for encoding complete thiamine biosynthesis

pathways. Variovorax was one of the genera with the greatest proportion of encoded thiamine biosynthesis. **c** The number of vitamins encoded by the 309 bioreactor-recovered genomes was assessed across genome length. Boxplot whiskers represent the minimum and maximum values within 1.5 times the inter-quartile range, while the bounds of the box indicate the 25th and 75th percentiles, and the center indicating the median. Genomes which encoded for the biosynthesis of 5-6 vitamins were statistically larger than those which encoded fewer (two-tailed independent samples *t*-test, $0.01 > p > 0.001$. where $n_{0\text{-}1} = 25$, $n_2 = 19$, $n_3 = 31$, $n_4 = 32$ and $n_{5\text{-}6} = 13$ biologically independent microbial genomes). Vitamins encoded vs vitamin encoded #2, *p*-value: 0-1 and 5-6 $p = 0.0014$; 2 and 5-6 $p = 0.0063$; 3 and 5-6 $p = 0.0051$; 4 and 5-6 $p = 0.0044$.

bacteria in the bioreactor communities are predicted to encode the genes necessary for thiamine synthesis. Unlike most prior studies that have used genomic data to investigate potential microbial inter-dependencies, we tested the prediction experimentally and correctly predicted thiamine production and interdependence. We confirmed vitamin exchange through isolation and characterization of the *Variovorax* predicted to be responsible for thiamine production, together with an isolated thiamine auxotroph, *Cryptococcus*. We report evidence that *Cryptococcus* is a pantothenic acid producer, a B-vitamin for which *Variovorax* is auxotrophic. This complementarity led to a 90–130-fold increase in cell density for both members when grown in co-culture. *Cryptococcus* was not present within the correlation network, likely due to its occurrence in only a handful of communities. Although it is possible that these two organisms are not directly interacting with each other in the mixed bioreactor communities, the exchange of vitamins between them in co-culture and the resulting fitness effects demonstrate that vitamin exchange is likely to have a potent impact on *Variovorax* and thiamine auxotrophs in the bior-eactor communities.

We noted a correlation between *Variovorax* and two *Saccharibacteria*, a *Kapabacteria*, a *Chloroflexi* and an *Arminatododetes* sp. (Fig. 4). The predicted *Microbacterium* host of one of the *Saccharibacteria* is also predicted to be a thiamine auxotroph. These findings raise the important possibility that the commonly predicted dependence of *Saccharibacteria* on an *Actinobacteria* species[28–31] can be more complex, in this case involving a third organism capable of vitamin production (*Variovorax*, which supplies thiamine). The significantly larger genome size of genomes, such as *Variovorax*, that encoded for more vitamin biosynthesis pathways suggests a tradeoff between genome streamlining and vitamin auxotrophies, as noted recently by Rodríguez-Gijón et al.[32]. This may be an important obser-vation for the identification of key vitamin providers in microbial communities.

*Variovorax* has been shown to be an important member of rhizo-sphere microbial communities[33]. Previously reported *Variovorax* gen-omes encode a repertoire of complex carbon compound degrading genes, which is consistent with the genetic potential of our *Variovorax* (Fig. 3). Members of this genus have been shown experimentally to

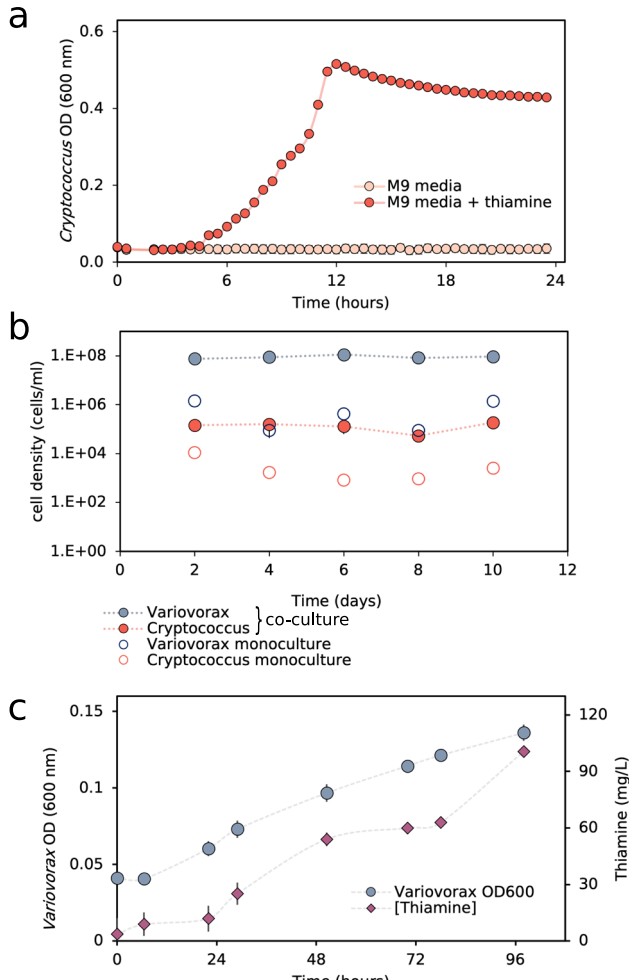

**Fig. 6 | *Variovorax* greatly enhances the fitness of the thiamine auxotroph *Cryptococcus* and produce large quantities of thiamine. a** A *Cryptococcus* isolated from the bioreactor community is shown to be a thiamine auxotroph, whereby its optical density (OD) in minimal media increases only with the addition of thiamine. **b** The steady-state cell density of this *Cryptococcus* increases ~50-fold when grown in co-culture with *Variovorax* in M9 media. *Variovorax* too receives a fitness advantage from the interaction. **c** *Variovorax* produces large quantities of extracellular thiamine in pure culture in minimal media supplemented with 25% spent media from *Cryptococcus* grown in minimal media for 5 days. Error bars represent one standard deviation from the mean of *n* = 3 biologically independent experiments.

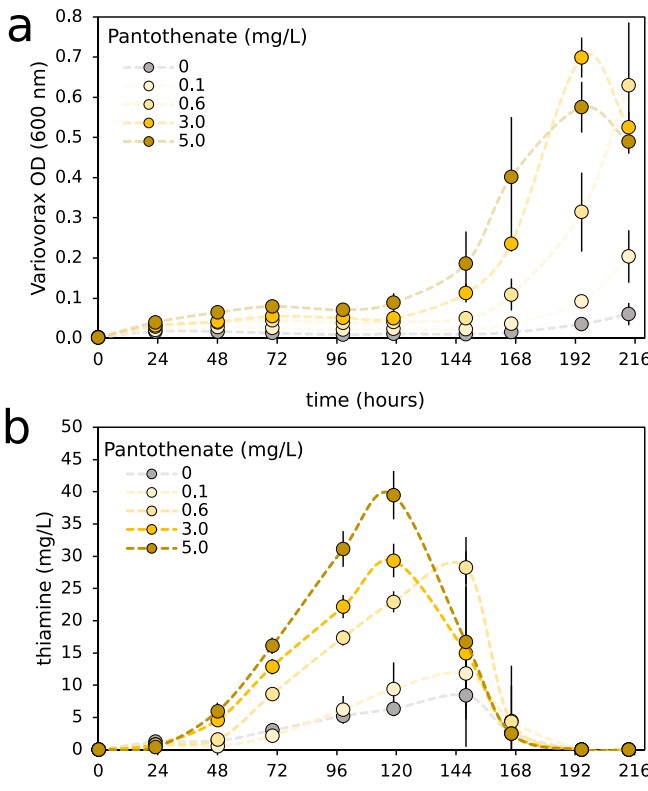

**Fig. 7 | *Variovorax* produces thiamine during lag-phase and consumes extracellular thiamine during exponential growth.** Growth, measured by optical density (OD), of *Variovorax* (**a**) and its thiamine production (**b**) when grown in M9 minimal media with the addition of varying concentrations of pantothenate (vitamin B5). Error bars represent one standard deviation from the mean of *n* = 4 biologically independent experiments.

degrade complex carbon compounds such as herbicides[34], and plant-produced isoprenes[35], the breakdown products of which support other microbial members unable to directly consume these compounds. Our results are the first to demonstrate *Variovorax* supports microbial communities through the provision of a vitamin. We found the thiamin biosynthesis operon is present in many members of the genus *Variovorax* (Fig. 5) and thus thiamine production may be a common function, in addition to its multiple other, already elucidated, roles.

*Variovorax* has been implicated in symbioses involving plants as well as bacteria. Finkel et al.[33] found that *Variovorax* was the sole member of a 185-member syncom responsible for the degradation of the bacteria-produced auxin which, unmetabolized, leads to root growth inhibition in *Arabidopsis* seedlings. *Variovorax* has also been noted frequently as a good plant endosymbiont[36]. An endophytic *Herbaspirrilum* was similarly found to produce a serobactin siderophore found in our *Variovorax* genome and was shown to directly aid in plant health under iron-limiting conditions[37]. These and other features support the potential of *Variovorax* as an agricultural probiotic[38,39].

We demonstrated that the *Variovorax* was not only capable of aiding the growth of a thiamine-auxotrophic microorganism but was able to produce extracellular concentrations of thiamine up to 100 mg/L. Bacteria typically require only 0.1–10 ng/ml for optimal growth[40,41] and natural and most engineered overproducers typically produce only 1–2 mg/L of extracellular thiamine[23,42,43]. However, thiamine deregulatory mutants have been developed for biotechnological application[23,44] that can produce up to 300 mg/L of thiamine. These mutations involve the deletion of the thiamine-monophosphate kinase (*thiL*) responsible for the phosphorylation of thiamine-phosphate, a thiamine permease (*yuaJ*) and active transporter responsible specifically for the influx of thiamine (*ykoD*). Analysis of the isolated *Variovorax* thiamine-monophosphate kinase (*thiL*) protein sequence appears functional based on a comparison with the crystal structure and sequence analysis performed by McCulloch et al.[24]. The active site residues of our *Variovorax's thiL* appear conserved and the additional conserved residues identified by McCulloch and co-authors were largely maintained. This supports that the regulation of this operon is functional. We suggest that *Variovorax* has evolved to produce large quantities of thiamine to support dependent organisms some of which, in turn, produce pantothenate. It is likely that cheaters, organisms that consume thiamine but do not produce anything beneficial to *Variovorax* in return, exploit this thiamine-producing phenotype. Overcoming cheaters may explain why such high concentrations of thiamine are produced by *Variovorax*.

The growth of *Variovorax* in minimal media could be stimulated through the addition of pantothenate. Supplementing greater concentrations of pantothenate led to a shorter duration of lag phase with higher supported cell densities. Thiamine overproduction was then

observed during this lag phase. The observed change from thiamine production to consumption by *Variovorax*, at the start of exponential growth, confirms that the thiamine biosynthesis operon can be properly regulated and is not constitutively-on as occurs in dysregulated mutants[44]. We hypothesize that the production of high concentrations of thiamine during lag phase may have evolved in this *Variovorax* to stimulate the growth of primarily pantothenate-producing bacteria. *Variovorax* is correlated with two pantothenate producers. Although *Crypotoccus* did not occur frequently enough to be included in the network, it probably exemplifies this co-dependency. The extreme thiamine-producing phenotype and the prevalence of this operon across the *Variovorax* genus suggest the potential of *Variovorax* for biotechnological vitamin production.

Recent studies simulating microbial communities with established microbial interactions have shown that inferring microbial interactions from correlations can be flawed, particularly, when correlations are drawn from noisy data[45]. The use of bioreactors to study microbial communities allows for many conditions to be maintained, such as temperature, dilution rate, and incoming solute concentrations. The steady-state conditions that are achieved rapidly in continuous stirred tank reactors leads to stable microbial communities and likely reduces noise and may have enabled more robust inference of microbial interaction from our network analysis. Our integration of network analysis with pathway reconstruction from genome-resolved metagenomics represents a readily implementable tool for identifying interacting microbial community members and their mechanism of interaction. The approach demonstrates the value of metagenomic datasets from sample series and is readily implementable in future studies of other microbial consortia.

In summary, using laboratory consortia experiments and cocultures of isolated microorganisms, we demonstrate the predictive power of combined network analyses, genome-resolved metagenomics, and pathway reconstruction for the elucidating interactions in complex microbial communities. We provide evidence for interdependencies such as between species of symbiotic *Saccharibacteria*, *Actinobacteria* and *Variovorax*, and for vitamin exchange, such as between *Variovorax* and *Cryptococcus*. Our results predict that *Variovorax* in our consortia has evolved to overproduce thiamine to support organisms that depend on it for this vitamin and provide evidence to suggest that thiamine production is likely an important trait for these bacteria in many ecosystems.

## Methods

### Bioreactor operation and sampling

The bioreactors used to study this microbial consortium were operated and sampled by Kantor et al.[9], Kantor et al.[10], Rahman et al.[11] and Huddy et al.[12]. Briefly, 1 L glass stirred-tank reactors were inoculated with homogenized biofilm and planktonic samples taken from long-running thiocyanate-degrading bioreactors at the University of Cape Town. The reactors were stirred using a pitched-blade impeller at 270 rpm and sparged with filtered air at 0.9 L/min. The bioreactors were fed continuously with deionized water containing 0.28 mM $KH_2PO_4$ and were buffered to pH 7.0 using 5 N KOH. Variable concentrations of sodium-thiocyanate (50–2000 mg/L) were supplied in the medium as the predominant electron donor. Molasses was included in some of the bioreactor mediums at 0.15 g/L while being omitted in others. All of the bioreactors were operated continuously at hydraulic retention times varying from 12–42 h. Biofilm and planktonic samples were recovered from the bioreactors at chemical steady states as described by the aforementioned studies.

### Microbial genomes and annotation

Previously published MAGs[9–12] recovered and reconstructed from 92 thiocyanate bioreactor metagenomes were pooled and a final genome set was produced using dRep (v3.0.0, https://github.com/MrOlm/drep)[46] at 95% ANI. Open reading frames (ORFs) were predicted using Prodigal v2.6.3[47,48] and annotated using Hmmscan (http://hmmer.org/) against KEGG (release 2022/04) and PFAM v35.0 databases. Vitamin pathways were recovered from KEGG website and applicable literature[49]. The capacity to produce a vitamin was determined based on a minimum set of genes representing different stages of each pathway. These minimum genes necessary for the deemed presence of a pathway are described in Supplementary data 5.

### Network and other statistical analysis

Reads were mapped to the dereplicated genome set, and relative abundance calculated, using CoverM (coverm genome -m relative abundance, https://github.com/wwood/CoverM). Coverm calculates the relative abundance of a genome by dividing the number of read bases that align with the genome by its total length. This value is then multiplied by the proportion of total reads mapped to the genome. Correlations, and associated *p*-values, between the genomes' relative abundance values were calculated using FastSpar[17] (https://github.com/scwatts/fastspar) which implements the SparCC algorithm[18] and again separately using Flashweave[19] (sensitive=true, heterogeneous=true, alpha=0.05, max_k = 0, https://github.com/meringlab/FlashWeave.jl).

Network analysis was performed using correlations greater or equal to 0.35. No negative correlations were considered. The Network analysis was performed using NetworkX[50] and was used to calculate the Betweenness-centrality (Eq. 1) for each node and the overall network density (Eq. 2).

$$b(v) = \Sigma(\sigma(s,t|v))/(\sigma(s,t)) \qquad (1)$$

Where σ (s, t) is the total number of shortest paths between nodes s and t, and σ (s, t | v)) is the number of those paths which pass through node v.

$$d = (2{*}m)/(n{*}(n-1)) \qquad (2)$$

Where m is equal to the number of edges and n is equal to the number of nodes.

A third metric was developed in this study for the quantification of the importance of a vitamin producer within the network (Eq. 3), termed the vitamin-hub metric.

$$h(v) = (D_{total}(v){*}D_{auxotroph}(v))/1/N{*}\Sigma(D_{auxotroph}, D_{producer}(v)) \qquad (3)$$

Where h is the vitamin-hub metric for node v, $D_{total}$ is the total number of edges of node v, $D_{auxotroph}$ is the number of edges from node v to vitamin auxotrophs, N is the total of these auxotrophic nodes, and Σ ($D_{auxotroph}$, $D_{producer}$) refers to summation of edges that these auxotrophic nodes share with nodes able to produce the vitamin.

This metric is calculated by multiplying the degree of a node, with the capacity to synthesize a given vitamin, by the number of edges it shares with nodes which are predicted to be auxotrophic for the given vitamin. This product is divided by the mean number of shared edges between each of the node's auxotrophic neighbors and nodes predicted to synthesize the vitamin. Networks were visualized using Cytoscape v3.8.2 (http://www.cytoscape.org/). Network modules were determined using the Louvain method implemented by Networkx[50]. Module detection was repeated with the Girvan-Newman algorithm (Fig. S2) and found to produce highly similar modules and were, therefore, not reported in the Results.

All comparisons of means and variances were conducted using a two-tailed independent samples *t*-test assuming unequal variance implemented in Python, utilizing the scipy.stats module from the SciPy library.

## GTDB *Variovorax* annotation and phylogenetic analyses

Species representatives in GTDB[51] (r207_v2; https://gtdb.ecogenomic.org/searches?s=al&q=Variovorax) were annotated as described above using KEGG HMMs and their capacity to synthesize thiamine evaluated. Genomes classified as *Variovorax* were then dereplicated at 98% average nucleotide identity (~subspecies level) using dRep[46]. GToTree v1.5.31[52] was used to identify, align, trim and concatenate 16 universal ribosomal proteins from the *Variovorax* genomes. A maximum-likelihood Phylogenetic reconstruction of these alignments was performed using IQtree[53] (-st AA -nt 48 -bb 1000 -m LG + G4 + FO + I). The phylogenetic tree was then annotated in iTol[54] v6.5.8. The contamination of these genomes was estimated using CheckM v1.2.1[55]. Complete thiamine biosynthesis pathways contained all KEGG annotations for (i) thiazole production, (ii) pyrimidine synthesis, (iii) the linking of these two moieties and (iv) the kinases involved in the overall biosynthesis process. Near-complete pathways were those that only lacked a phosphomethylpyrimidine synthase (*thiC*, K03147) from this pathway.

## Microbial isolation, sequencing and assembly

Microbial isolates were recovered from an in-house thiocyanate-degrading bioreactor by serial dilution in sterile 1x PBS buffer and spread plating onto R2A solid agar medium. Recovered isolates were grown in liquid R2A medium and 600 uL combined at 1:1 with 50% (v/v) glycerol before being stored at −80 °C. The remaining cell culture was pelleted at 10 000 g for 10 min and used for subsequent DNA extraction. Total DNA was extracted from the isolates using a DNAeasy® PowerSoil® Pro DNA extraction kit (Machery-Nagel, Germany). Paired-end Illumina TruSeq libraries with fragment sizes of 500 bp were sequenced on an Illumina NovaSeq to yield 250 bp paired-end reads. Reads were trimmed using Sickle (www.github.com/najoshi/sickle) and assembled using IDBA-UD[56] using default settings. Annotation was performed as described above. Genome assemblies were manually inspected for contamination and completeness based on the GC, coverage and phylogeny of scaffolds and the presence of bacterial single copy marker genes using ggKbase (https://ggkbase.berkeley.edu).

## Isolate culturing and co-culturing

The recovered isolates were screened for thiamine auxotrophies using solid M9 minimal media (0.4 g/L glucose, 2 mM MgSO$_4$, 0.1 mM NaCl, 0.048 mM Na$_2$HPO$_4$.7H$_2$O, 0.022 mM KH$_2$PO$_4$, 0.017 mM NaCl, 15 g/L agar), with and without the addition of 0.5 mg/L thiamine. The *Cryptococcus* species identified as a possible thiamine auxotroph was grown in a 96-well plate in liquid M9 minimal media with ($n = 6$) and without ($n = 6$) the addition of 0.5 mg/L thiamine. Growth was quantified spectrophotometrically at 600 nm.

To evaluate the effect of the growth of *Variovorax* on *Cryptococcus*, and vice versa, each of these isolates were grown in pure culture and in co-culture in minimal media. Initially, the isolates were grown in R2A medium overnight, spun down at 3000 g for 3 min and washed in M9 minimal media before being inoculated into 10 ml M9 minimal media each at an initial OD$_{600}$ of 0.035. The pure and co-cultures were allowed to grow for 48 h before passaging into fresh M9 media, and this repeated for a total of 4 passages. Immediately following each passage, viable cell densities were determined by serial dilution in M9 media and spread plating onto R2A media agar. These plates were incubated at 30 °C for 48 h and colonies were counted and differentiated based on colony morphology (*Cryptococcus* white, *Variovorax* yellow). Only biological and no technical repeats were performed in these experiments.

*Variovorax* was cultured in M9 minimal media in the presence of varying concentrations of pantothenate using the methodology described above but allowed to grow for a total of 9 days. The cultures were performed with four replicates and were sampled aseptically every 24 h and quantified for cell optical density. Subsequently, cells

were pelleted at 10 000 g for 3 min in a benchtop centrifuge and filter sterilized using a 0.22 um filter. These solutions were then used for the quantification of extracellular thiamine.

## Thiamine and pantothenic acid quantification

The quantity of thiamine produced by Variovorax was quantified during the growth in 75% fresh minimal media and 25% filter-sterilized spent medium from the final passage of the co-culture experiment as well as during the growth of Variovorax in M9 minimal media with varying concentrations of pantothenate. Thiamine was quantified on an Agilent 1200 liquid chromatograph equipped with a Agilent ZORBAX Eclipse Plus C-18 column with a 5 um particle size, maintained at 35 °C. Using a two-phase mobile phase of (A) 25 mM NaH$_2$PO$_4$ (pH = 2.5), and (B) pure methanol. The flow rate was 1.0 mL/min. The mixture of these mobile phases were as follows: 0% mobile phase B from time of injection, 50% mobile phase B from 1.0 min to 75% mobile phase B from 10 to 25 min. We identified vitamins using a G1362A Refractive Index Detector at 220 nm. Vitamin standards were obtained from Sigma-Aldrich®.

## Reporting summary

Further information on research design is available in the Nature Portfolio Reporting Summary linked to this article.

## Data availability

Variovorax and Proteobacterial genomes studied for pathway completeness were recovered from GTDB (r207_v2, https://gtdb.ecogenomic.org/). The 309 dereplicated bioreactor genomes used in this study are available through the online database ggKbase at https://ggkbase.berkeley.edu/SCN_92/organisms. The raw read data and all MAGs from the previous studies and used in this study are available at NCBI (http://www.ncbi.nlm.nih.gov/) under accession codes PRJEB29350 (https://www.ncbi.nlm.nih.gov/bioproject/513540), PRJNA279279 and PRJNA629336. Source data are provided with this paper.

## Code availability

Custom code used to perform network analyses and calculate vitamin hub metrics (doi: 10.5281/zenodo.8122228) can be found at https://github.com/tomas-hesler/Vitamin_interdependencies_2023/.

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

## Acknowledgements

This material by m-CAFEs Microbial Community Analysis and Functional Evaluation in Soils, (m-CAFEs@lbl.gov) a Science Focus Area led by Lawrence Berkeley National Laboratory is based upon work supported by the U.S. Department of Energy, Office of Science, Office of Biological and Environmental Research under contract number DE-AC02-05CH11231. We would like to acknowledge Dr Christine He, Dr Sumayah Rahman and Dr Rose Kantor sample collection and the processing of metagenomic sequencing data. We thank Yi Liu for assistance with HPLC, Dr Maria Lukarska for assistance with *thiL* sequence comparison, and to Prof Michi Taga for valuable discussions.

## Author contributions

T.H., S.D and J.F.B. conceived the study and designed the experiments. T.H. performed informatic analysis and conducted the experiments. T.H. and J.F.B. wrote the initial manuscript. R.J.H, S.L, R.S., S.T.L.H. contributed critical resources and advice. All authors contributed to the manuscript.

## Competing interests

J.F.B. is a co-founder of Metagenomi. The remaining authors declare no competing interests.
