## [Peer Review File · Nature Communications]

REVIEWER COMMENTS

Reviewer #1 (Remarks to the Author):

This manuscript presents an analysis of potential vitamin cross-feeding in microbial communities with a focus on thiamine production. A co-occurrence network analysis was performed on a preexisting time-resolved metagenomic dataset, and observations were validated using a *Variovorax* isolate identified as a hub in this network in co culture with a *Cryptococcus* isolate. While the authors make some interesting observations, there are several issues with the study that should be addressed.

The use of correlation-based network analysis in microbiome data has been a subject of debate for the past two decades, with challenges posed by the compositionality of the datasets and the difficulty of distinguishing similar niche preferences from interactions. It is not entirely clear how these challenges were addressed in the study, and whether the results are affected by these issues. The authors use linear correlations to analyze the relationship between vitamin-producing organisms and their potential consumers, but it is not clear why this approach was chosen given its sensitivity to compositionality. The authors acknowledge (Line 392) that non-linear relationships should be considered as well. I would advocate for attempting this within the current manuscript, at least in order to see whether the *Variovorax*'s "hubness" is retained across network construction methods.

Additionally, the interpretation of the network analysis results is somewhat vague. The authors describe the network as "sparse" based on a value of 6.7 percent, but provide no context for this claim. Similarly, a modularity value of 0.52 is reported without context.

The authors mention (Line 300) that non-thiamine producing *Variovorax* mutants have been studied. Using such a mutant is needed here to substantiate that *Cryptococcus* growth is indeed increasing 90 fold due to *Variovorax* derived thiamine and not due to some other service provided by *Variovorax*. This seems like a low hanging fruit that I'm surprised was not picked.

Figure-text coherence: ensure that all graphs of the figures are referenced in the text in order.

Abbreviations should be referenced before using them. For example – "B1" in Figure 2.

Line 155: Figure 2B is not referred to anywhere in the text (and neither is 2C). It appears from this figure that network hubs do not necessarily produce more vitamins than non-hub nodes. There seem to be

plenty non hub vitamin B producers as well, which raises the question of whether similar co-culture results would be observed with a non-hub Thiamine producer. Additionally, the authors note that *Cryptococcus* was omitted from the network since it was only found in several samples. But it is not reported whether *Variovorax* is enriched in the communities where *Cryptococcus* was detected. In general, these issues indicate a certain disconnect between the use of network analysis and the observation that vitamin crossfeeding is important.

Line 167: S4 was called earlier. Why not switch the order of these 2 figures?

Line 180: The value was not found to be significantly different from the mean, this does not mean that enrichment is partial. It means that the data does not support the hypothesis that there is enrichment at all.

Line 202: this only includes known biosynthetic genes. There is a possibility that there are additional as of yet unannotated ones. I suggest adding the word “known” to the sentence.

Line 261: Please show this data.

Figure S2 and elsewhere – axis titles are tiny.

Figure S3: the wording of the explanation for panel B is somewhat unclear.

Reviewer #2 (Remarks to the Author):

The authors used 92 previously collected metagenomic samples of bioreactors run in chemostat mode to compute co-occurrence networks with SparCC for 309 MAGs. After discarding negative edges, eight hub nodes were identified. *Variovorax* was the only hub node for which a corresponding species was isolated. Its corresponding network module was found to be enriched in thiamin auxotrophs; vitamin sharing by module members was subsequently experimentally confirmed. It is important to experimentally validate edges in co-occurrence networks. However, I have some concerns as detailed below.

There are many co-occurrence network construction approaches available. The authors selected SparCC; do the hub nodes and modules change when another, more recent, network construction tool is used, such as FlashWeave or SPIEC-EASI?

The authors mention biofilm samples. Did these give co-occurrence networks comparable to planktonic samples?

The authors merged several chemostat experiments. Were the bioreactors run with the same media and settings, i.e. were batch effects not an issue, or did the authors construct networks separately for each data set?

L. 442-443: "We suggest that *Variovorax* has evolved to produce large quantities of thiamine to support dependent organisms some of which, in turn, produce pantothenate."

How does *Variovorax* escape cheaters, i.e. organisms that use thiamin but do not share pantothenate in return?

How about testing the enrichment of the *Variovorax* module in thiamin auxotrophs with Fisher's exact test/hypergeometric distribution? Then no simulations are required.

L. 123: "We used positive linear correlations"

Correlation is able to handle some types of non-linearity. As long as values are monotonically increasing/decreasing, correlation measures do not need values to be perfectly linear to be significant.

The authors should comment on the importance of using bioreactors to control confounding factors and what it means for the application of their method to field samples.

Reviewer #3 (Remarks to the Author):

The manuscript "Vitamin interdependencies predicted by metagenomics-informed network analyses validated in microbial community microcosms" reports vitamin cross-feeding interactions in a complex microbial community. Cross-feeding was first predicted based on genome annotation-based metabolic pathway analysis and the construction of abundance correlation networks. A particular strength of the study is that targeted co-culture experiments validated the predicted metabolic complementary and

cross-feeding. The metagenome-based and network-oriented methodology is potentially of great interest to many researchers in microbial ecology (although it needs to be presented in more detail in the manuscript; see below), as the same workflows could be adapted to other microbial communities. The manuscript is well-written, and the authors provide access to the underlying metagenomic data.

Some suggestions for the authors to consider in their revisions:

The description for the construction of abundance correlation networks requires more information. What exact numbers were correlated in the abundance correlation networks? The methods section states that metagenomic reads were mapped to dereplicated MAGs, but there is no information on how the mapping statistics were used as a proxy for relative abundance. Were read counts normalised by genome size? Or was the average/median coverage per MAG used as a proxy for abundance? These details can have a strong influence on the correlation results.

The definition of an auxotrophy in an organism X has two requirements: (1) X is not able to synthesise the focal metabolite de-novo, and (2) the focal metabolite is essential for the growth of X. Not all vitamins are universally essential for growth in prokaryotes (<https://doi.org/10.1016/j.ymben.2016.12.002>). Therefore, this should ideally also be mentioned in the manuscript when defining vitamin auxotrophies in the present genotypes.

The study relies on variation in the community composition of the same microbial community under different environmental conditions to have data for the abundance correlation analysis. The authors report that the varying environmental conditions included different concentrations of thiocyanate. I suggest the authors provide more details on the other varying conditions that resulted in different microbial microbe abundances in the community. Moreover, it should be discussed if one would expect different network analysis results if other varying growth conditions had been chosen.

Minor point: Line 496: "using Prodigal's" -> "using Prodigal".

Response to reviewers

Reviewer #1

*This manuscript presents an analysis of potential vitamin cross-feeding in microbial communities with a focus on thiamine production. A co-occurrence network analysis was performed a preexisting time-resolved metagenomic dataset, and observations were validated using a *Variovorax* isolate identified as a hub in this network in co-culture with a *Cryptococcus* isolate. While the authors make some interesting observations, there several issues with the study that should be addressed.*

The use of correlation-based network analysis in microbiome data has been a subject of debate for the past two decades, with challenges posed by the compositionality of the datasets and the difficulty of distinguishing similar niche preferences from interactions. It is not entirely clear how these challenges were addressed in the study, and whether the results are affected by these issues.

The authors were aware of the potential drawbacks of correlation-based networks. We used Fastsparr which implements the sparCC algorithm, which was developed to handle compositional data, and also ran analyses using standard Spearman correlation on relative abundance data. SparCC incorporates a sparsity constraint that reduces the number of correlations estimated which reduces the risk of false discoveries as compared to spearman correlation using compositional data. SparCC generated far fewer and presumably more robust correlations. We

concede that a proportion of the correlations giving rise to the edges between nodes likely resulted from two organisms simply occupying the same niche. To better separate niche preferences from interactions we identified 'network hubs' using betweenness centrality instead of degree alone, so as to favor connectivity within the entire graph. Such hubs would be unlikely to arise from similar niche preferences as many organisms would have to be involved. We now clarify the use of Fastsparr/SparCC to address the challenge of compositional data in the results and methodology sections.

The authors use linear correlations to analyze the relationship between vitamin-producing organisms and their potential consumers, but it is not clear why this approach was chosen given its sensitivity to compositionality. The authors acknowledge (Line 392) that non-linear relationships should be considered as well. I would advocate for attempting this within the current manuscript, at least in order to see whether the Variovorax's "hubness" is retained across network construction methods.

As noted above, we used SparCC to minimize the effects of compositional data. We note that reviewer 3 points out the wrong-mindedness of Line 392, so we have deleted that statement. The accurate prediction seems to suggest that linear correlation using SparCC was sufficient to detect relationships between Variovorax and thiamine-dependant organisms.

Additionally, the interpretation of the network analysis results is somewhat vague. The authors describe the network as "sparse" based on a value of 6.7 percent, but provide no context for this claim. Similarly, a modularity value of 0.52 is reported without context.

We thank the reviewer for this comment. To justify the interpretation of "sparse" we have added context by noting that a value of 100% would have every node connected to every other node. Similarly to provide context for the modularity value of 0.52 we have provided a reference.

The authors mention (Line 300) that non-thiamine-producing Variovorax mutants have been studied. Using such a mutant is needed here to substantiate that Cryptococcus growth is indeed increasing 90-fold due to Variovorax-derived thiamine and not due to some other service provided by Variovorax. This seems like a low-hanging fruit that I'm surprised was not picked.

We believe this may have been a misunderstanding. We have clarified in the text that it was Variovorax mutants that were engineered to overproduce thiamine that were studied (genes required for the proper regulation of the operon were mutated).

Figure-text coherence: ensure that all graphs of the figures are referenced in the text in

order. Thank you for noting this. This has been corrected.

Abbreviations should be referenced before using them. For example – “B1” in Figure 2.

“Vitamin B1” has now been introduced in line 163 ahead of Figure 2. We did not find any other abbreviations that were not defined.

Line 155: Figure 2B is not referred to anywhere in the text (and neither is 2C).

Thank you for pointing this out. These two subplots are now referred to in the text:

(i) It appears from this figure that network hubs do not necessarily produce more vitamins than non-hub nodes.

We agree. We sought to determine whether any of the hub’s importance could be explained by its ability to produce a vitamin, but we did not try to rule out the possibility of abundant vitamin production by other organisms. We have clarified this in the Results.

“It is worth noting that the metrics used in the analysis do not indicate that the identified organisms are the only ones in their communities capable of producing the specific vitamins. Instead, the purpose is to identify potential vitamin producers that may be relied upon by other auxotrophs in the community.”

(ii) There seem to be plenty non-hub vitamin B producers as well, which raises the question of whether similar co-culture results would be observed with a non-hub Thiamine producer.

The authors agree that other co-cultures could be generated between, for example, *Cryptococcus*, and other vitamin-producing organisms represented in our network but which are not predicted hubs. However, experiments involving randomly chosen organism pairs would not contribute to the main objective of this study, which was to find evidence for vitamin sharing involving an organism (*Variovorax* in this case) that is a hub in a microbial community interaction network.

(iii) Additionally, the authors note that Cryptococcus was omitted from the network since it was only found in several samples. But it is not reported whether Variovorax is enriched in the communities where Cryptococcus was detected. In general, these issues indicate a certain disconnect between the use of network analysis and the observation that vitamin crossfeeding is important.

We have not found evidence for *Variovorax* enrichment in the few communities where *Cryptococcus* was detected, but organism abundance is the consequence of many factors and so we do not assign significance to this result.

We feel that the novelty of this study is in part due to the combination of microbiome cultivation under a wide range of conditions, network analysis, and hypothesis testing using isolates. Regarding the seeming disconnect between the network analysis and the vitamin-sharing component, *Variovorax* was chosen as the focus of our work based on the network topology rather than via analysis of pairwise interactions. The network was used to deduce that vitamin supply underlies *Variovorax*'s hub status by mapping metagenomically establishing the lack of thiamine biosynthetic capacity in organisms linked to *Variovorax* in the network. We then support this inference by showing that *Variovorax* is capable of providing enough thiamine to support multiple organisms. We have clarified this in the Discussion.

Line 167: S4 was called earlier. Why not switch the order of these 2 figures?

Figure S3 and Figure S4 have been switched.

Line 180: The value was not found to be significantly different from the mean, this does not mean that enrichment is partial. It means that the data does not support the hypothesis that there is enrichment at all.

We have updated the sentence in Line 180 to read: "Therefore, *Variovorax*'s high thiamine hubness is not a result of auxotroph enrichment in its local network neighborhood, but instead being due to *Variovorax* being one of the few thiamine producers with which the thiamine auxotrophs are correlated."

Line 202: this only includes known biosynthetic genes. There is a possibility that there are additional as of yet unannotated ones. I suggest adding the word "known" to the sentence.

We agree with this and have updated the sentence to include "known".

Line 261: Please show this data.

A boxplot showing the relative abundance of the *Saccharibacteria* when *Variovorax* is below and above 0.15 % relative abundance has been incorporated into Figure S6 (previously Figure S5).

Figure S2 and elsewhere – axis titles are tiny.

Figure font sizes have been increased throughout the manuscript.

Figure S3: the wording of the explanation for panel B is somewhat unclear.

This now reads:“(B) The proportion of each of the 92 microbial communities that encode for a complete thiamine biosynthesis pathway.”

Reviewer #2

The authors used 92 previously collected metagenomic samples of bioreactors run in chemostat mode to compute co-occurrence networks with SparCC for 309 MAGs. After discarding negative edges, eight hub nodes were identified. Variovorax was the only hub node for which a corresponding species was isolated. Its corresponding network module was found to be enriched in thiamin auxotrophs; vitamin sharing by module members was subsequently experimentally confirmed. It is important to experimentally validate edges in co-occurrence networks.

We thank this reviewer for their careful reading of the manuscript and appreciation of our experimental approach to validate network-based predictions.

However, I have some concerns as detailed below.

There are many co-occurrence network construction approaches available. The authors selected SparCC; do the hub nodes and modules change when another, more recent, network construction tool is used, such as FlashWeave or SPIEC-EASI?

We thank this reviewer for the suggestion. We agreed that it would be interesting to test one other tool and have conducted analyses using FlashWeave, as suggested. We find that the network structure does show some change depending on which correlation method is used. However, we note that Variovorax remains one of the highest-scoring thiamine-hubs across both methodologies. The following has been included in the results:

“We employed Flashweave (Tackmann et al., 2019) to explore the possibility of detecting the predicted Variovorax phenotype using an alternative correlation method (Figure S5). The resulting network exhibited a higher number of predicted correlations compared to SparCC (Fastspar) and displayed modular characteristics, consisting of six submodules with a modularity score of 0.497. Out of the 164 genomes in this network, 49 were predicted to be capable of producing thiamine. We calculated the thiamine hub metric for these nodes and we observed that the majority of the top-scoring nodes in this network coincided with the top-scoring nodes identified in the SparCC network. These included the Comamonas (Node 261), Bradyrhizobiaceae (Node 233), Nitrospira

(Node 176) and importantly, *Variovorax* (Node 55). Interestingly, the *Comamonas* (node 261) predicted to be a network hub from the SparCC network had the highest betweenness centrality of any node in the Fastweave network.”

The authors mention biofilm samples. Did these give co-occurrence networks comparable to planktonic samples?

The biofilms and planktonic samples (which derive from the same reactor) share a large fraction of organisms. We chose to include both sets of samples because to reject either the biofilm or planktonic samples results in a loss of the statistical power needed for our analyses.

The authors merged several chemostat experiments. Were the bioreactors run with the same media and settings, i.e. were batch effects not an issue, or did the authors construct networks separately for each data set?

Each of the many reactors had the same working volume, was maintained at room temperature, was fed the same base media composition, and was inoculated with a microbial community originating from a single environmental community. The incorporation of molasses, the thiocyanate concentration and the dilution rate applied to these reactors varied. This is discussed in more detail in the Results section.

The datasets were not split before network analysis as this would severely limit the number of samples used to construct each network. Batch effects were not investigated, however, the study uses data generated from a large number of bioreactors.

*L. 442-443: “We suggest that *Variovorax* has evolved to produce large quantities of thiamine to support dependent organisms some of which, in turn, produce pantothenate.” How does *Variovorax* escape cheaters, i.e. organisms that use thiamin but do not share pantothenate in return?*

This is an important point and we thank the reviewer for raising this. We suspect cheaters do exploit this phenotype of *Variovorax*. Only two direct correlations are observed between *Variovorax* and a pantothenate producer, both of which are thiamine auxotrophs. The possible exploitation by cheaters may provide some explanation as to why *Variovorax* produces so much thiamine.

We have discussed this further following this sentence in line 442-443: “It is likely that ‘cheaters’, organisms that consume thiamine but do not produce anything beneficial to *Variovorax* in return, exploit this thiamine-producing phenotype. Overcoming cheaters may explain why such high concentrations of thiamine are produced by *Variovorax*.”

How about testing the enrichment of the Variovorax module in thiamin auxotrophs with Fisher's exact test/hypergeometric distribution? Then no simulations are required.

We considered doing this analysis but we were concerned that the small sample size may give rise to misleading results. Further, we wanted to go beyond enriched versus not enriched (which is what these tests would tell us) so we used the simulation to understand the basis of *Variovorax*'s high hubness score. We find that the distribution of simulated scores provides more information and context to the result.

L. 123: "We used positive linear correlations" Correlation is able to handle some types of non-linearity. As long as values are monotonically increasing/decreasing, correlation measures do not need values to be perfectly linear to be significant.

This sentence is in a section where we discuss the drawbacks to the approach we used. As we agree with the reviewer on this point we have deleted this sentence from the Discussion.

The authors should comment on the importance of using bioreactors to control confounding factors and what it means for the application of their method to field samples.

This is an excellent point. We have included the following paragraph in the discussion section.

"The use of bioreactors to study microbial communities allows for many conditions to be maintained, such as temperature, dilution rate, and incoming solute concentrations. The steady-state conditions that are achieved rapidly in continuous stirred tank reactors leads to stable microbial communities which is important for robust network analysis."

Reviewer #3

The manuscript "Vitamin interdependencies predicted by metagenomics -informed network analyses validated in microbial community microcosms" reports vitamin cross-feeding interactions in a complex microbial community. Cross-feeding was first predicted based on genome annotation-based metabolic pathway analysis and the construction of abundance correlation networks. A particular strength of the study is that targeted co-culture experiments validated the predicted metabolic complementary and cross-feeding. The metagenome-based and network-oriented methodology is potentially of great interest to many researchers in microbial ecology (although it needs to be presented in more detail in the manuscript; see below), as the same workflows could be adapted to other microbial communities. The manuscript is well-written, and the authors provide access to the underlying metagenomic data.

We thank this reviewer for their careful reading and positive comments.

The description for the construction of abundance correlation networks requires more information. What exact numbers were correlated in the abundance correlation networks? The methods section states that metagenomic reads were mapped to dereplicated MAGs, but there is no information on how the mapping statistics were used as a proxy for relative abundance. Were read counts normalized by genome size? Or was the average/median coverage per MAG used as a proxy for abundance? These details can have a strong influence on the correlation results.

The following section in the methodology (from Ln 504) has been revised to:

“Reads were mapped to the dereplicated genome set, and relative abundance calculated, using CoverM (coverm genome -m relative abundance, <https://github.com/wwood/CoverM>). Coverm calculates the relative abundance of a genome by dividing the number of read bases that align with the genome by its total length. This value is then multiplied by the proportion of total reads mapped to the genome. Correlations, and associated p-values, between the genomes’ relative abundance values were calculated using FastSpar (Watts et al., 2019) which implements the SparCC algorithm (Friedman & Alm, 2012).”

The definition of an auxotrophy in an organism X has two requirements: (1) X is not able to synthesise the focal metabolite de-novo, and (2) the focal metabolite is essential for the growth of X. Not all vitamins are universally essential for growth in prokaryotes (<https://doi.org/10.1016/j.ymben.2016.12.002>). Therefore, this should ideally also be mentioned in the manuscript when defining vitamin auxotrophies in the present genotypes.

This is an important point and we are grateful to the reviewer for raising this. To address this we have made multiple changes. The following sentence has been added to the introduction:

“Not all bacteria need certain vitamins. A vitamin auxotroph is therefore defined as an organism (i) unable to synthesize its own vitamin (ii) which it requires for growth. However, Xavier et al. (2017) predicted some vitamins and cofactors as essential in bacteria, including thiamine, coenzyme-A (for which pantothenic acid is required for biosynthesis) and NAD(H).”

The following sentence has been added to the Results section with an associated supplemental figure:

“We found that 99% of the 309 genomes analyzed in this study encoded enzymes that are essential for the TCA cycle, as shown in Figure S4. However, three *Saccharibacteria* genomes were an exception as they rely on their host for these genes and metabolites (He et al., 2015). Additionally, we found that each genome, including the *Saccharibacteria*, contained a minimum of two and a median of eight enzymes that require thiamine as a cofactor, as shown in Figure S4.”

The study relies on variation in the community composition of the same microbial community under different environmental conditions to have data for the abundance correlation analysis. The authors report that the varying environmental conditions included different concentrations of thiocyanate. I suggest the authors provide more details on the other varying conditions that resulted in different microbial microbe abundances in the community. Moreover, it should be discussed if one would expect different network analysis results if other varying growth conditions had been chosen.

Each of the many reactors had the same working volume, was maintained at room temperature, was fed the same base media composition, and was inoculated with a microbial community originating from a single environmental community. The incorporation of molasses, the thiocyanate concentration and the dilution rate applied to these reactors varied. This is discussed in more detail in the Results section and details can be found in (Kantor et al., 2015; 2017; Rahman et al., 2017; Huddy et al., 2021).

The following has been included in the Discussion section:

In future work, it would be interesting to explore how higher concentrations of molasses, other complex substrates, or vitamin additives would reconfigure the network.

Minor point: Line 496: "using Prodigal's" -> "using Prodigal".

This has been corrected in text.

Additional changes

Original figures 3 and 4 were condensed into a single figure.

Additional data were incorporated into Figure 5 - thiamin proportions from Proteobacteria GTDB and the genome size vs number of vitamins produced.

REVIEWERS' COMMENTS

Reviewer #1 (Remarks to the Author):

My comments were mostly addressed. I do have one remaining issue. I apologize for my misunderstanding of the nature of the existing Thiamine mutants. Nevertheless, my point was that *Variovorax* appear to be amenable to genetic manipulation. Genetic proof that this operon is indeed the cause of the increase in *Cryptococcus* growth would greatly strengthen this paper in my view. This comments still stands despite my error, and I would appreciate it if the authors addressed it.

Reviewer #2 (Remarks to the Author):

The authors have addressed a number of my comments, but I still have a few issues with this manuscript.

"The biofilms and planktonic samples (which derive from the same reactor) share a large fraction of organisms. We chose to include both sets of samples because to reject either the biofilm or planktonic samples results in a loss of the statistical power needed for our analyses."

That means that many edges in the network will be driven by differences between biofilm and planktonic microbial composition, which needs to be discussed.

Given the large number of network clustering algorithms available (e.g. MCL, MCODE, WGCNA, manta, Girvan-Newman etc), please motivate the use of the Louvain network clustering algorithm. If Louvain was an arbitrary choice, then check whether algorithm choice matters.

"The steady-state conditions that are achieved rapidly in continuous stirred tank reactors leads to stable microbial communities which is important for robust network analysis"

Why is a steady state important to microbial network inference? It is definitely possible to infer microbial networks from transient dynamics (e.g. <https://www.nature.com/articles/nature13828>).

To make results reproducible, the networks and the abundance tables from which they were constructed should be made publicly available. Also, in the future, include metadata (such as dilution rate or biofilm/planktonic sample status) in the network to explore the impact of varying conditions

Reviewer #3 (Remarks to the Author):

The authors have addressed all my comments and suggestions from the first revision round.

Response to reviewers

July 7th 2023

Vitamin interdependencies predicted by metagenomics-informed network analyses and validated in microbial community microcosms

Reviewer #1 (Remarks to the Author):

My comments were mostly addressed. I do have one remaining issue. I apologize for my misunderstanding of the nature of the existing Thiamine mutants. Nevertheless, my point was that Variovorax appear to be amenable to genetic manipulation. Genetic proof that this operon is indeed the cause of the increase in Cryptococcus growth would greatly strengthen this paper in my view. This comments still stands despite my error, and I would appreciate it if the authors addressed it.

We appreciate that the proposed experiment would be interesting, but there are significant challenges involved which we feel have been addressed through alternate experiments in the paper. The greatest challenge is that under the conditions required for the co-culture experiment (e.g. no thiamine present in the medium) knocking out the thiamine operon in *Variovorax* would abolish its growth. This mutant would therefore not be capable of supporting a co-culture with *Cryptococcus*. Alternatively, it was shown that the addition of thiamine alone enabled the growth of *Cryptococcus* (Figure 6A) and it was demonstrated that *Variovorax* is capable of producing an unprecedented concentration of thiamine (>100 mg/L, Figure 6C).

Reviewer #2 (Remarks to the Author):

"The biofilms and planktonic samples (which derive from the same reactor) share a large fraction of organisms. We chose to include both sets of samples because to reject either the biofilm or planktonic samples results in a loss of the statistical power needed for our analyses."

That means that many edges in the network will be driven by differences between biofilm and planktonic microbial composition, which needs to be discussed.

We agree with this comment and have added additional analyses relating to the relative abundance and the frequency of organisms in biofilm and planktonic communities as well as from samples that received high (>250 mg/L) and low (≤250 mg/L) thiocyanate concentrations. These data have been overlaid onto the network and are summarized in Figures S2 and S3. We have included the following paragraph in the results section:

"We investigated how environmental conditions may affect the network structure. We categorized the samples into biofilm or planktonic as well as environments with high or low thiocyanate

concentrations. Most network hubs showed similar frequency and abundance in biofilm and planktonic samples (Figure S2 and S3). However, the Chitinophagaceae (Node 221) was more frequent and abundant in planktonic communities as well as high thiocyanate samples. This was true for the other nodes in this node's module. In contrast, the module associated with the Alphaproteobacteria hub (Node 54) was more frequent and abundant in low thiocyanate samples and in planktonic communities. The other determined hub nodes showed little difference between these conditions, suggesting they have a broader ecological tolerance within these bioreactor systems."

Given the large number of network clustering algorithms available (e.g. MCL, MCODE, WGCNA, manta, Girvan-Newman etc), please motivate the use of the Louvain network clustering algorithm. If Louvain was an arbitrary choice, then check whether algorithm choice matters.

The Louvain algorithm is commonly used in microbial ecology to define network modules. We had not performed module detection with other methods and based on this comment we repeated this with the Girvan-Newman algorithm. This method too is widely used and distinct from the Louvain algorithm. Girvan-Newman produced modules highly similar to those produced using Louvain and would therefore not change the interpretation of the network. We have updated Figure S1 to show modules determined using both methods.

"The steady-state conditions that are achieved rapidly in continuous stirred tank reactors leads to stable microbial communities which is important for robust network analysis"
Why is a steady state important to microbial network inference? It is definitely possible to infer microbial networks from transient dynamics (e.g. <https://www.nature.com/articles/nature13828>).

We thank the reviewer for this comment and agree that our statement requires clarification. It is true that networks can be generated from transient dynamics. However, recent studies have shown that inferring microbial interactions from correlations can be flawed under several conditions. In particular, data that contains considerable noise is the most susceptible to incorrectly inferring interactions from correlations, and incorrectly classifying the type of interaction (e.g. competition). Therefore, we speculate that sampling at steady states is one approach to reduce noise and better infer microbial interactions. (<https://journals.plos.org/ploscompbiol/article?id=10.1371/journal.pcbi.1010491>, <https://www.nature.com/articles/s41396-019-0459-z>) We have revised this section to:

"A recent study simulating microbial communities with established microbial interactions has shown that inferring microbial interactions from correlations can be flawed, particularly, when correlations are drawn from noisy data (Pinto et al., 2022). The use of bioreactors to study microbial communities allows for many conditions to be maintained, such as temperature, dilution rate, and incoming solute concentrations. The steady-state conditions that are achieved rapidly in continuous stirred tank reactors lead to stable microbial communities and limit transient

dynamics. The data generated under these conditions may have enabled a more robust inference of microbial interaction from our network analysis.”

To make results reproducible, the networks and the abundance tables from which they were constructed should be made publicly available. Also, in the future, include metadata (such as dilution rate or biofilm/planktonic sample status) in the network to explore the impact of varying conditions.

We have made the abundance table, detailed node information, and the network edge table in the supplementary data. We have now updated this with metadata relating to biofilm and planktonic samples and other operating conditions.